# Vision Language Model Based Caption Evaluation Method Leveraging Visual Context Extraction

## Abstract

Given the accelerating progress of vision and language modeling, accurate evaluation of machine-generated image captions remains critical. In order to evaluate captions more closely to human preferences, metrics need to discriminate between captions of varying quality and content. However, conventional metrics fall short of comparing beyond superficial matches of words or embedding similarities; thus, they still need improvement. This paper presents VisCE$^2$, a vision language model-based caption evaluation method. Our method focuses on visual context, which refers to the detailed content of images, including objects, attributes, and relationships. By extracting and organizing them into a structured format, we replace the human-written references with visual contexts and help VLMs better understand the image, enhancing evaluation performance. Through meta-evaluation on multiple datasets, we validated that VisCE$^2$ outperforms the conventional pre-trained metrics in capturing caption quality and demonstrates superior consistency with human judgment.

## 1 Introduction

The evaluation of the machine-generated image caption is a core research topic to illustrate models' ability to describe their visual observation in textual forms and shape the branch of vision and language modeling studies into meaningful and grounded directions. In the early stage of neural image captioning research, such as neural image caption generator (Vinyals et al., 2015), attention-based (Xu et al., 2015), and sentinel and spatial attention (Lu et al., 2017), models had enabled more and more detailed and accurate descriptions. Hence, they had achieved better and better performance on reference-based automatic evaluation metrics such as BLEU (Papineni et al., 2002) and CIDEr (Vedantam et al., 2015). SPICE (Anderson et al., 2016) was also proposed to assess better correspondences of reference and generated captions in grammatical aspects. More recently, BERTScore (Zhang et al., 2020) and CLIPScore (Hessel et al., 2021) have been introduced to measure the similarity between the embeddings of generated captions and references.

In very recent advances in vision and language models (VLMs), however, model generations become so detailed that they often exceed the capability of the automatic evaluation metrics and even the entire coverage of annotated references. Both InstructBLIP (Dai et al., 2023) and LLaVA (Liu et al., 2023b) follow textual instructions and generate tailored descriptions that are not even similar to references but are of high quality. Considering the great advancements of these recent models, we go back to the basics of image captioning to tailor new evaluation metrics: first capturing the contents of the images and then rearranging them to describe and composing a phrase. In assessing image captioning abilities, they correspond to different aspects: coverage of image contents, accuracy in describing them, and compositional sophistication. In this line of evaluation metrics, InfoMetIC (Hu et al., 2023) relies on matching image regions and words in the captions. While it is quite sensitive to the alignments of objects in images and captions, it becomes less sensitive to supportive facts such as attributes or interactions of objects. Unfortunately, they are also learning-based metrics and depend on fine-tuning with multiple captioning datasets.

This paper concentrates on reference-free image caption evaluation and proposes a new **Vis**ion Language Model-based **C**aption **E**valuation Method leveraging **Vis**ual **C**ontext **E**xtraction (**VisCE**$^2$). In

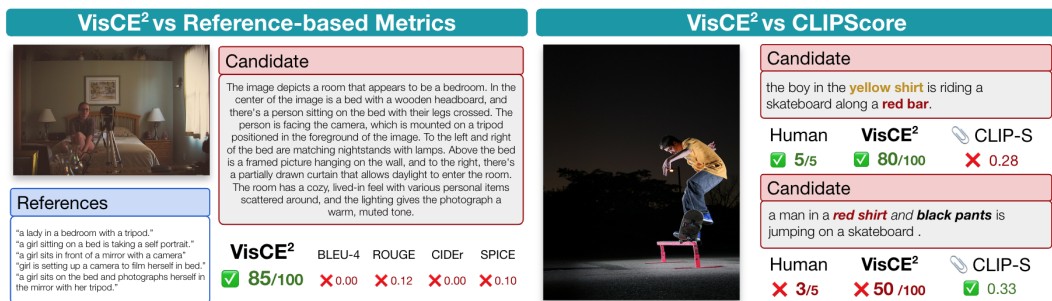

Figure 1: Comparison between conventional methods and our reference-free method, **VisCE$^2$**. Our method not only evaluates longer captions more accurately compared to reference-based methods (left) but also assesses compositional errors in captions more effectively than CLIP-S (right).

this context, "visual context" refers to information about objects, attributes, and their relationships, including those in the background and inconspicuous objects. This visual context is given to the model in the evaluation process with an image and a candidate caption. Explicitly providing the visual context in a structured format, rather than reference images or hand-written captions, helps the VLM better understand the images' content and expose how accurately the candidate caption describes the parts of the image or which parts are missing.

The comparison in Figure 1 emphasizes the effectiveness of VisCE$^2$ over conventional evaluation methods. The left panel presents a caption generated by GPT-4 using the straightforward prompt, *"Generate a detailed description for the given image"* and scores of evaluation metrics. This caption accurately describes the image's content in quite a detailed manner. Indeed, its detailedness overwhelms human-written references, and hence, all of the reference-based metrics undervalued the caption. The right panel highlights that CLIP-S fails to detect compositional errors, such as color misidentifications (*"**red shirt** and **black pants**"*), resulting in high scores for inaccurate description. Conversely, VisCE$^2$ effectively identifies these discrepancies, demonstrating its superior capability in evaluating image captions.

We investigated the quality of the proposed evaluation method on several image caption datasets, THumB (Kasai et al., 2022), Flickr8k-Expert (Hodosh et al., 2013), Composite (Aditya et al., 2015), and Pascal-50S (Vedantam et al., 2015). Our method, VisCE$^2$, outperformed conventional metrics and correlated highly with human judgments. Furthermore, meta-evaluation experiments uncovered that the evaluation scores of VisCE$^2$ strongly correlate with the accuracy of candidate captions. Through a series of exhaustive ablation experiments, we verified that the proposed method is effective with state-of-the-art VLMs. Moreover, we have quantitatively shown that using larger LMs in VisCE$^2$ improves the evaluation performance.

## 2 RELATED WORK

### 2.1 EVALUATION METHOD FOR IMAGE CAPTIONING

**Text-only (reference-based) methods**. Image captioning has been evaluated using a combination of several metrics. While some have been adapted from metrics used in other NLP tasks such as machine translation and summarization (BLEU (Papineni et al., 2002), METEOR (Denkowski & Lavie, 2014), ROUGE (Lin, 2004)) and others proposed for image captioning (CIDEr (Vedantam et al., 2015), SPICE (Anderson et al., 2016)), all of them are mainly based on $n$-gram matches with the reference caption. Following these classical approaches, an evaluation metric was proposed that exploits the versatility of pre-trained models: BERTScore (Zhang et al., 2020) measures the similarity of embeddings output by BERT (Devlin et al., 2018) for each reference and candidate caption. In BERTScore++ (Yi et al., 2020), they fine-tuned BERT to the task using the image caption dataset. More recently, CLAIR (Chan et al., 2023) utilized large language models for evaluation. This ensembles ChatGPT (OpenAI, 2022), Claude (Bai et al., 2022), and PaLM (Chowdhery et al., 2022) and achieves high evaluation performance by providing only reference sentences and instructions for

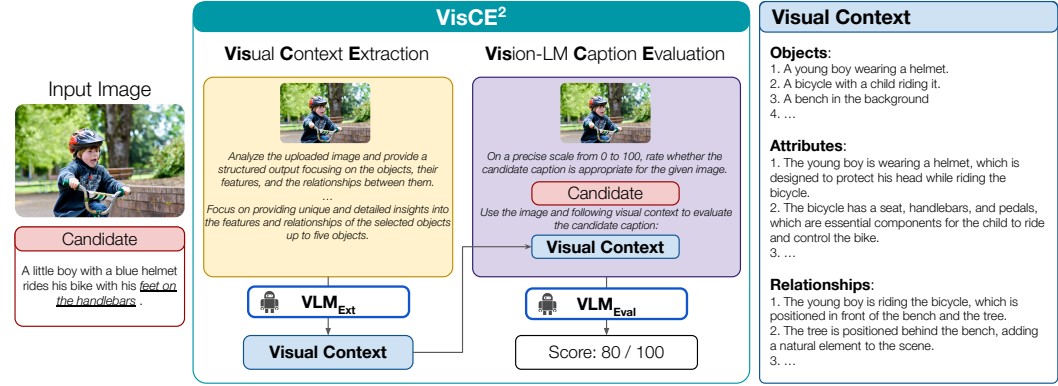

Figure 2: Overview of automatic caption quality evaluation by VisCE$^2$ and an example of the input/output. First, VLM extracts the visual context from the image, organized in a bullet list format, presenting objects, object attributes, and relationships between objects. Then, VLM evaluates the caption using the obtained visual context along with the image and candidate caption.

the task. However, these metrics are sensitive to the quality and coverage of the reference captions available.

**Crossmodal methods**. Covering the rich visual information in an image with only pre-defined reference captions is difficult. To alleviate this loss of information, evaluation methods based on vision and language models (VLMs), which leverage features of the images, have been proposed. TIGEr (Jiang et al., 2019) uses a pre-trained SCAN model (Lee et al., 2018), fine-tuned on the COCO dataset (Chen et al., 2015), and calculates how much the candidate caption is grounded in the image. ViLBERTScore (Lee et al., 2020) extracts features of the images and calculates the score as with BERTScore with a pre-trained ViLBERT model (Lu et al., 2019). FAIEr (Wang et al., 2021) connects images and texts via scene graphs and computes scores according to their overlap.

**Image-only (reference-free) methods**. To simultaneously improve evaluation performance and reduce the cost of annotation for references, several methods have been proposed for image caption evaluation. CLIP-S (Hessel et al., 2021) attaches the score by simply calculating the modified cosine similarity between embeddings of an image and that of a candidate caption using CLIP (Radford et al., 2021). PAC-S (Sarto et al., 2023) refined the pre-training method of CLIP with data augmentation using an image captioner (Li et al., 2022) and image generator (Rombach et al., 2022), resulting in improved evaluation performance. Other methods employ quality estimators for evaluation. UMIC (Lee et al., 2021) fine-tunes UNITER (Chen et al., 2020) via contrastive loss for gold and automatically perturbated caption pair. InfoMetIC (Hu et al., 2023) fuses image and language modalities by stacking CLIP and Transformer (Vaswani et al., 2017), fine-tuned with large image-caption datasets (Young et al., 2014; Aditya et al., 2015).

Unlike these methods, our method utilizes visual context to detail the structure of the image content for caption evaluation. In our evaluation protocol, the VLM extracts the visual context to encapsulate the comprehensive image content and feeds it to the VLM itself. By doing so, it is possible to refer to information from both the vision and language sides, and hence, it is expected to improve the accuracy of quality estimation.

## 2.2 RECENT VISION-LANGUAGE MODELS

The recent vision and language model proposal has progressed the fusion understanding of image and language modalities. One of the most important milestones is CLIP (Radford et al., 2021). Contrastive learning on a large-scale image-text dataset constructed by web crawling has significantly improved zero-shot performance on various vision and language tasks. Following this, the development of various models accelerated progress in the V&L domain. BLIP (Li et al., 2022) refined pre-trained methods by enhancing the quality of image and text data by the image captioner. OFA (Wang et al., 2022) integrated various V&L tasks by modifying the input-output architecture. Several other studies also presented remarkable performance, which are still being upgraded (Bai

et al., 2023; Zhu et al., 2023; Chen et al., 2023; Li et al., 2023). In addition, InstructBLIP (Dai et al., 2023) and LLaVA (Liu et al., 2023b) allow us to perform various tailored tasks according to instructions. Furthermore, state-of-the-art VLMs such as GPT-4V (OpenAI et al., 2023) and Gemini (Google, 2023) are now available via APIs.

We focus on the fidelity to instructions and hence applied our VisCE$^2$ to the LLaVA-v1.5 (Liu et al., 2023a) to ensure transparency and reproducibility in our experiments.

## 3 PROPOSED METHOD: VISCE$^2$

We introduce a reference-free automatic caption evaluation method, **VisCE$^2$**. Figure 2 shows the overview of VisCE$^2$. Our method takes an image and a caption as inputs and predicts a rating score for how accurately the caption describes the image. This assessment is conducted via two key components: visual context extraction and VLM-based caption evaluation.

**Visual Context Extraction**. VisCE$^2$ initially extracts a detailed visual context from an input image using VLM, named VLM$_{\text{Ext}}$. We define *visual context* as the content of an image classified into objects, object attributes (including color, shape, and size), and relationships between objects, following the similar notion of scene graphs (Xu et al., 2017). To extract relevant image details, we instruct the VLM$_{\text{Ext}}$ to articulate the visual context in a bullet list format by category rather than employing the structured graphical representation due to its complexity and the difficulties associated with textual representation. We believe that providing the successive VLM$_{\text{Eval}}$ module with detailed visual context in a structured format helps bridge the gap between the reference image and the candidate text. This facilitates the model's evaluation both consistently and comprehensively.

**Vision-LM Caption Evaluation**. In the next step of VisCE$^2$, VLM$_{\text{Eval}}$ evaluates the candidate caption based on the extracted visual context and an input image. As with many other LLM-based tasks, VisCE$^2$ treats caption evaluation as a multimodal text completion task. The VLM$_{\text{Eval}}$ is given a prompt combining a reference image, visual context, and a candidate caption as input and then generates an output sentence that encapsulates the overall quality scores ranging from 0 to 100. In the preliminary experiments, we found that some VLMs occasionally disregard instructions to generate only evaluation scores, instead producing sentences that include these scores (e.g., *"The score is X out of 100"*.) In the postprocessing phase, we eliminated these canonical phrases and designated the first integer value in the output sentences as the evaluation score. In contrast to previous embedding-based methods, such as CLIP-S (Hessel et al., 2021), which cannot determine the quality of scores without multiple examples, our approach can provide absolute scores close to human intuition, allowing for determining the caption's quality by just looking at a single caption.

## 4 EXPERIMENT

This section demonstrates the effectiveness of our VisCE$^2$ and conducts a meta-evaluation of the automatic evaluation methods.

### 4.1 EXPERIMENTAL SETTINGS

**Implementation details**. To ensure the transparency and reproducibility of experimental results, we use `LLaVA-v1.5-13B` as the base model, one of the best-performing models among the publicly available VLMs with the default hyperparameter settings, and report the results of a single run. We set the max token length to 1,024 in extracting visual context. Note that VisCE$^2$ allows for the use of any VLM without restrictions. Hence, we also employed five other models in the ablation studies ( Sec. 5.1) to confirm the model-agnostic effectiveness.

**Baseline metrics**. We compared the evaluation performance against five of the most common metrics in the automatic evaluation of image captioning: BLEU (Papineni et al., 2002), ROUGE (Lin, 2004), METEOR (Denkowski & Lavie, 2014), CIDEr (Vedantam et al., 2015), and SPICE (Anderson et al., 2016). These metrics evaluate the overlap of *n*-grams with reference captions. Specifically, SPICE generates semantic scene graphs from the candidate captions using dependency parse trees, assessing matches based on objects, attributes, and their relationships. Furthermore, we employed more recent reference-based measure BERTScore++ (Yi et al., 2020) and modern reference-free

Table 1: Correlation (Pearson's $\rho$) between baseline metrics and human judgment on THumB 1.0. "w/o" means discarding human annotated samples. **Bold** fonts for best score among **reference-free** and **tune-free** models. †: The scores reported in previous works.

| Method | Reference-free | Tune-free | THumB w/o Human | | | THumB w/ Human | | |
|---|---|---|---|---|---|---|---|---|
| | | | P | R | Total | P | R | Total |
| BLEU-4 | ✗ | ✓ | .21 | .13 | .25 | .15 | .04 | .13 |
| ROUGE | ✗ | ✓ | .26 | .17 | .31 | .18 | .07 | .18 |
| CIDEr | ✗ | ✓ | .27 | .18 | .33 | .21 | .11 | .23 |
| SPICE | ✗ | ✓ | .26 | .15 | .30 | .20 | .09 | .21 |
| RefCLIP-S | ✗ | ✓ | .34 | .27 | .44 | .31 | .26 | .41 |
| †InfoMetIC | ✓ | ✗ | .22 | .30 | .37 | .21 | .32 | .38 |
| CLIP-S | ✓ | ✓ | .18 | **.27** | .32 | .17 | **.28** | .32 |
| **VisCE$^2$ (Ours)** | ✓ | ✓ | **.54** | .08 | **.45** | **.49** | .06 | **.39** |

metrics, CLIP-S (Hessel et al., 2021), and PAC-S (Sarto et al., 2023). Among reference-free metrics, InfoMetIC (Hu et al., 2023) relies on a quality estimator that is fine-tuned with the training splits of Flickr-30k (Young et al., 2014) and MS-COCO (Chen et al., 2015). In contrast, the other metrics, including our VisCE$^2$, do not depend on such data-specific fine-tuning. Furthermore, CLAIR (Chan et al., 2023) exploits the zero-shot evaluation results of proprietary LLMs such as ChatGPT, Claude, and PaLM. While we include their results for comparison in the following experiments, it should be noted that they are not directly comparable without careful consideration.

**Evaluation datasets**. We conducted a meta-evaluation of automatic evaluation metrics across four image captioning datasets: THumB 1.0 (Kasai et al., 2022), Flickr8k-Expert (Hodosh et al., 2013), Composite (Aditya et al., 2015), and Pascal-50S (Vedantam et al., 2015). We provide detailed descriptions of the datasets in Appendix A.

**Meta-evaluation metrics**. Following previous studies, we utilized three different indicators which correspond to each evaluation dataset: Pearson's correlation coefficient $\rho$ for measuring the linear correlation between two sets of data, Kendall's $\tau$ for measuring the ordinal association between two measured quantities, and the accuracy as of the percentage of the correct pairwise ranking between two candidate captions.

## 4.2 CORRELATION WITH HUMAN JUDGMENT

We analyze the proposed VisCE$^2$ metrics on the THumB1.0 (Kasai et al., 2022) dataset by assessing the correlation between automatic evaluation scores and human ratings in three aspects: precision, recall, and the total score. Following previous studies, we used Pearson's correlation coefficient as the meta-evaluation index.

The results we presented in Table 1 highlight VisCE$^2$'s outstanding performance in terms of correlation with precision, surpassing all other metrics. It exceeded RefCLIP-S, the previous state-of-the-art metric, by 0.20 and 0.18 points in settings with and without human-written captions, respectively. This suggests that VisCE$^2$ accurately evaluated precise captions. On the other hand, VisCE$^2$'s performance in recall presented a contrast, exhibiting minimal correlation.

In the THumB dataset, recall scores reflect the extent to which the caption encompasses the salient information in the image. Meanwhile, the evaluation scores of our method were calculated by considering all objects, attributes, and their relationships within the image. The difference between focusing on the salient objects and on all objects may have led to the low correlation with recall scores. Despite its imbalanced nature, our method is highly correlated with the total scores, indicating the overall quality of captions.

We also test the caption evaluation capability of VisCE$^2$ using Flickr8k-Expert (Hodosh et al., 2013) and Composite (Aditya et al., 2015) dataset. Kendall's rank correlation coefficient was used as a meta-evaluation index to measure the correlation between automatic and human evaluation.

VisCE$^2$ performed at a high level compared to the other automatic evaluation metrics in its correlation with human evaluation (Table 2 left). VisCE$^2$ in the Flickr8k-Expert achieved state-of-the-art

Table 2: Correlation between human judgement and metrics on Flickr8k-Expert and Composite, and accuracy on Pascal-50S.

| Method | Reference-free | Tune-free | Flickr8k-Expert Kendall's $\tau$ | Composite Kendall's $\tau$ | PASCAL-50S HC | HI | HM | MM | Mean |
|---|---|---|---|---|---|---|---|---|---|
| BLEU-4 | ✗ | ✓ | 30.6 | 28.3 | 53.0 | 92.4 | 86.7 | 59.4 | 72.8 |
| ROUGE | ✗ | ✓ | 32.1 | 30.0 | 51.5 | 94.5 | 92.5 | 57.7 | 74.0 |
| METEOR | ✗ | ✓ | 41.5 | 36.0 | 56.7 | 97.6 | 94.2 | 63.4 | 77.9 |
| CIDEr | ✗ | ✓ | 43.6 | 34.9 | 53.0 | 98.0 | 91.5 | 64.5 | 76.7 |
| SPICE | ✗ | ✓ | 51.7 | 38.8 | 52.6 | 93.9 | 83.6 | 48.1 | 69.5 |
| †BERTScore++ | ✗ | ✗ | 48.1 | 42.3 | 65.4 | 98.1 | 96.4 | 60.3 | 80.1 |
| RefCLIP-S | ✗ | ✓ | 52.6 | 51.2 | 64.9 | 99.5 | 95.5 | 73.3 | 83.3 |
| †RefPAC-S | ✗ | ✓ | 55.5 | 51.5 | 67.7 | 99.6 | 96.0 | 75.6 | 84.7 |
| †CLAIR$_{\text{Claude}}$ | ✗ | ✓ | 56.2 | 54.2 | 57.9 | 98.5 | 91.3 | 62.9 | 77.6 |
| †CLAIR$_{\text{E}}$ | ✗ | ✓ | 62.7 | 59.2 | 57.7 | 99.8 | 94.6 | 75.6 | 81.9 |
| †InfoMetIC | ✓ | ✗ | 54.2 | 59.2 | 69.0 | 99.8 | 94.0 | 78.3 | 85.3 |
| CLIP-S | ✓ | ✓ | 51.1 | 49.8 | 55.9 | 99.3 | 96.5 | 72.0 | 80.9 |
| †PAC-S | ✓ | ✓ | 53.9 | .51.5 | 60.6 | 99.3 | **96.9** | **72.9** | **82.4** |
| **VisCE$^2$ (Ours)** | ✓ | ✓ | **59.0** | **55.0** | 60.7 | 99.6 | 93.6 | 69.3 | 80.8 |

performance by a significant margin of more than 4.8 pts compared to other reference-free metrics, demonstrating that our method is highly capable of estimating the quality of image captions aligned with human judgments. A similar trend was observed in the Composite. Compared to CLIP-S and PAC-S, VisCE$^2$ surpassed them by over 3pts. It was found that our method fell short of InfoMetIC's performance. This degradation is assumed to originate from the training method of InfoMetIC, a fine-tuning quality estimator on the three datasets that comprise the Composite dataset. The result also offers insight into the quality estimation based on large-scale models. While CLAIR$_{\text{E}}$, which utilizes an ensemble of non-public LLMs such as ChatGPT, PaLM, and Claude, achieved significant results in the benchmarks, VisCE$^2$ outperformed the outcome of CLAIR$_{\text{Claude}}$, which solely exploits Claude. This distinction highlights the efficacy of our method, considering its reliance on publicly available and comparatively smaller models. Such performance showcases the potential of more accessible models to achieve high-quality benchmarks.

To further verify how accurately our VisCE$^2$ determines the relative preference of captions, we conducted a meta-evaluation with the Pascal-50S dataset (Vedantam et al., 2015) that comprises human preference judgments indicating which of the two captions for an image is more appropriate. The preference judgments for pairs of candidate captions are classified into the following four categories;

1. HC: pairs of human-written captions, both of which correctly represent the image's content;
2. HI: pairs of human-written captions where one caption is correct and the other is incorrect;
3. HM: pairs of correct captions, one of which is human-written and the other machine-generated;
4. MM: pairs of machine-generated captions, both of which correctly represent the image's content.

For each category, we measured accuracy, the judgment agreement between the human preferences, and the results of the automatic evaluation method.

The pairwise ranking agreement result shown in the right side of Table 2 depicts that our method outperformed CLIP-S and PAC-S in the HC and HI categories. On the other hand, relatively low performance was observed in HM and MM compared to other metrics. This degraded performance, especially noticeable for MM, can be attributed to the nature of the candidate captions. The automatically generated captions in the Pascal-50S dataset are made with classical captioning methods, such as Midge (Mitchell et al., 2012) and Babytalk (Kulkarni et al., 2011), often containing errors. Therefore, both are judged as poor-quality captions, causing the model's performance to deteriorate. Another contributing factor might be the disagreement in objectives between human annotation and automatic quality estimation. In the annotation process for the Pascal-50S dataset, the worker selected captions that are more similar to randomly selected references, which does not necessarily equate to superiority or adequacy as a caption.

Table 3: The result of ablation studies. Correlations with human ratings are measured by Pearson's $\rho$, Kendall's $\tau$, and agreement with human preference by accuracy.

| | THumB w/o human | | | THumB w/ human | | | Flickr8k-Exp. | Composite | Pascal-50S |
| | Pearson's $\rho$ | | | Pearson's $\rho$ | | | Kendall's $\tau$ | Kendall's $\tau$ | Accuracy (%) |
| | P | R | Tot. | P | R | Tot. | | | Mean |
|---|---|---|---|---|---|---|---|---|---|
| **VisCE$^2$ (Ours)** | **.54** | .08 | **.45** | **.49** | .07 | **.39** | **59.0** | **56.0** | **80.8** |
| I: **Visual Context** | | | | | | | | | |
| None (Vanilla) | .47 | .08 | .38 | .46 | .09 | .36 | 55.9 | 52.4 | 80.5 |
| w/ References | .41 | .06 | .32 | .38 | .07 | .30 | 54.6 | 54.5 | 79.2 |
| w/ Description | .50 | .08 | .41 | .48 | .09 | .38 | 57.4 | 52.5 | 77.5 |
| II: **Component on Visual Context** | | | | | | | | | |
| VisCE$^2$ w/o Rel. , Attr. | .51 | .08 | .41 | .48 | .08 | .37 | 55.8 | 55.2 | 80.4 |
| VisCE$^2$ w/o Rel. | .53 | .07 | .43 | .49 | .07 | .38 | 56.2 | 55.7 | 80.5 |
| III: **VLM$_{Ext}$ & VLM$_{Eval}$** | | | | | | | | | |
| LLaVA-v1.5-7B | .23 | .04 | .20 | .22 | .05 | .18 | 29.0 | 34.2 | 73.1 |
| LLaVA-v1.6-vicuna-7B | .27 | .08 | .22 | .25 | .08 | .20 | 24.7 | 35.7 | 70.0 |
| LLaVA-v1.6-vicuna-13B | .55 | .06 | .43 | .50 | .06 | .37 | 55.8 | 52.7 | 75.5 |
| mplug-owl2-llama2-7B | .44 | .13 | .38 | .41 | .13 | .35 | 43.7 | 50.8 | 77.4 |
| GPT-4o | .54 | .16 | .49 | .52 | .19 | .47 | 54.5 | 59.0 | 83.0 |
| IV: **VLM$_{Ext}$** | | | | | | | | | |
| Smaller (LLaVA-v1.5-7B) | .46 | .11 | .40 | .43 | .12 | .36 | 55.2 | 53.9 | 76.6 |
| Larger (GPT-4o) | .43 | .13 | .40 | .40 | .12 | .35 | 55.1 | 54.3 | 76.6 |

## 5 ANALYSIS

### 5.1 ABLATION STUDIES

We conducted ablation studies to explore which parts of VisCE$^2$ improve evaluation performance. The results of ablation studies are listed in Table 3.

**I: Effectiveness of Visual Context**. We initially investigated the impact of incorporating visual context into the auto-evaluation method. In addition to the candidate captions and images as VLM input, we compared the following four types of text: (i) *Vanilla* uses only the task instruction, (ii) *w/ Reference* attaches references provided within datasets, (iii) *w/ Description* detailed captions generated by LLaVA-1.5-13B, and (iv) *VisCE$^2$ (Ours)* leverages visual contexts extracted during the initial step of VisCE$^2$. See Appendix B for the prompts.

The top of Table 3 presents the results for different visual contexts for VLM$_{Eval}$. Our VisCE$^2$ demonstrated superior evaluation performance across all datasets, suggesting that incorporating visual context enhances overall evaluation performance, except recall. In contrast, other methods, such as *w/ Reference* and *w/ Description*, showed no improvement or even degradation from the *Vanilla* on some datasets. This suggests that the visual context's structured and comprehensive nature enables consistent caption evaluations.

To further analyze VisCE$^2$, we compared the distribution of scores assigned to captions on three different datasets in the Vanilla and VisCE$^2$ settings. The heatmap presented in Figure 3 plots the distribution of human and automatic evaluation scores in THumB, Flickr8k-Expert, and Composite, which comprise Likert scale human evaluations. Heatmaps are normalized by human rating points to remove sample size bias in the data set. Thus, each row sums to 1, and the heatmap rows represent the ratio of the automatic rating score to a caption of a specific human rating. We observed that the score distribution in the *Vanilla* setting has two prominent peaks, with captions classified as good (70+) and bad (0-10), whereas VisCE$^2$ exhibits three distant peaks: good (70-80), bad (0-10), and fair (40-50). Furthermore, in THumB and Composite, the VisCE$^2$ heatmap shows a decrease in the value of the upper right corner when compared to the Vanilla heatmap, indicating a reduction in the percentage of overrated captions. These distribution changes suggest that introducing visual context contributes to a closer evaluation of the human impression.

**II: Effect of Each Component on Visual Context**. After validating the effectiveness of visual context, we subsequently compared the settings that feed VLM$_{Eval}$ a partially clipped visual context with VisCE$^2$ to investigate which parts of the visual context contribute to superior performance. We compared the performance of VisCE$^2$ with the following settings: (i) *w/o Relation, Attribute*

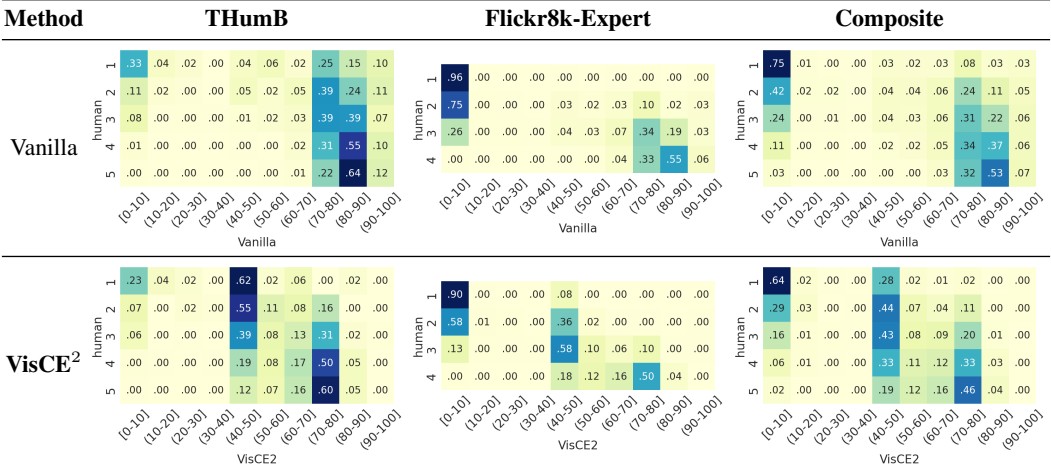

Figure 3: Heatmaps of human rating and automatic evaluation scores on THumB (left), Flickr8k-Expert (mid) and Composite (right). Normalized for each human evaluation score (i.e., rows). The human evaluation of THumB is referenced to the total score.

uses only object information, and (ii) *w/o Relation* employs object and attribute information. As demonstrated in the middle of Table 3, the performance of VisCE$^2$ improves with the addition of more visual context components. Experimental results validated that providing a detailed and well-organized visual context enables consistent caption evaluations by offering richer information for the VLM$_{\text{Eval}}$. This supports the hypothesis that a more detailed visual context improves evaluation performance.

**III: Effect of Backbone VLM**. We compared several backbone models to examine whether performance differences depend on the variant or model size. We employed five models from current strong VLMs (LLaVA-v1.5-7/13B (Liu et al., 2023a), LLaVA-v1.6-7/13B (Liu et al., 2024), mPLUG-Owl2 (Ye et al., 2023)) as for open-source models (see Appendix C for model details.) We used the same backbone model for both VLM$_{\text{Ext}}$ and VLM$_{\text{Eval}}$. The results showed that models with larger language model sizes consistently exhibited higher scores than their smaller counterparts, regardless of the dataset. Furthermore, there was no significant difference in performance between LLaVA-v1.5-13B (our standard model) and LLaVA-v1.6-vicuna-13B. Our analysis reveals that the evaluation performance is dependent on the LM size. However, model updates do not always yield positive impacts.

We also explored the current upper bounds of VisCE$^2$ and the effectiveness of proprietary models using GPT-4o. Our preliminary experiment (see Appendix D.1) revealed that GPT-4 with VisCE$^2$ has a higher evaluation performance than LLaVA-v1.5. We used the `gpt-4o` model through the Azure OpenAI API. The experimental result shows that GPT-4o surpassed those of open models on almost all datasets. On THumB, Pearson's $\rho$ with human ratings increased by 8 points, and Kendall's $\tau$ on Composite by 3 points. This result suggests that the proposed method is effective even with more advanced VLMs. While VisCE$^2$ with GPT-4o is a reference-free and tuning-free evaluation method, its performance is comparable to the state-of-the-art reference-based or fine-tuned metrics.

**IV: Effect of VLM$_{\text{Ext}}$**. To examine whether the performance of VisCE$^2$ is affected by the inference model, we meta-evaluated VisCE$^2$ with different VLM$_{\text{Ext}}$. We fixed the VLM$_{\text{Eval}}$ to LLaVA-v1.5-13B and changed VLM$_{\text{Ext}}$ into a smaller model (LLaVA-v1.5-7B) and a larger model (GPT-4o). In the bottom of Table 3, we observed that the performance decreased in both cases when the VLM$_{\text{Ext}}$ was changed. This finding implies that the performance of VisCE$^2$ is improved by effectively executing in-context learning using the same model.

## 5.2 QUALITATIVE ANALYSIS

To clarify the differences in trends among other reference-free metrics, we compared the evaluation scores assigned to the images by VisCE$^2$ with those of CLIP-S, a typical reference-free metric.

| Image | Candidate Caption | Human | VisCE$^2$ | CLIP-S |
|---|---|---|---|---|
| | a group of people on a field playing baseball. | 5 | 85 | 0.31 |
| | a baseball player swinging a bat at a ball . | 1 | 0 | 0.26 |
| | The scene contains people wear hats and greenery and people wear helmets and people wear sports dresses and crowd. | 3 | 50 | 0.20 |
| | a man in orange garb carrying a umbrella and cell phone. | 5 | 80 | 0.41 |
| | a woman holding an umbrella in the rain . | 1 | 0 | 0.31 |
| | The scene contains street and people walk and booths and cars arranged in some fashion and cars. | 2 | 50 | 0.23 |
| | A woman sitting at a table with a vase of food. | 3 | 50 | 0.32 |
| | A woman sitting at a table in a restaurant. | 4.5 | 80 | 0.32 |
| | A woman sits beside a brick wall at a small table in a restaurant. | 4.4 | 70 | 0.32 |
| | a dog leaps out of the water. | 3.0 | 70 | 0.34 |
| | a dog is running through the water . | 5.0 | 85 | 0.33 |

Figure 4: Comparison between evaluation scores of VisCE$^2$, that of CLIP-S, and human ratings for candidate caption for images from Composite and THumB dataset. 1 to 5 are the human ratings, where 5 is the best. Additional examples are provided in the Appendix D.2.

Figure 4 presents examples of human ratings and scores of the auto-eval methods for image-caption pairs. These qualitative examples indicate the discriminative ability of VisCE$^2$ to the accuracy of captions. CLIP-S tends to overestimate captions describing the presence of objects in the image. For example, CLIP-S assigns a relatively high score to the incorrect caption *"a baseball player swinging a bat at a ball."* to the image in the first row. However, although "the baseball player" is present in the image, the event of "swinging the bat at the ball" has not occurred. Humans can detect such contradictions accurately, whereas CLIP-S often ignores them. In the second example, CLIP-S also overestimated incorrect image descriptions containing the salient object, an umbrella, in the image. A previous study (Ahmadi & Agrawal, 2023) has also pointed out that CLIP-S tends to be affected by the presence or absence of descriptions of salient objects in the image, confirmed in these examples.

The third and fourth examples are typical cases where VisCE$^2$ correctly distinguishes appropriate image descriptions. These examples demonstrate that VisCE$^2$ can accurately evaluate captions based on the image content, even when the differences between captions are subtle. In particular, in the fourth example, VisCE$^2$ correctly evaluates the image by understanding the captions *"leaps out"* and *"run through"* and provides a human-like evaluation. In contrast, CLIP-S cannot provide significantly different evaluations for similar captions due to its nature of calculating the similarity between embeddings. These differences in scores are a significant feature of VisCE$^2$, which provides evaluations closer to human ratings.

## 6 CONCLUSION

We have proposed VisCE$^2$, a prompting method for automatic VLM-based evaluation of image captions. VisCE$^2$ deviates from the traditional method of evaluating similarity using only reference text and images, weaving visual context into the evaluation framework with a compositional form. This technique allows the VLMs to understand the detailed visual dependencies of images better and validate them based on more exhaustive content than the reference captions provide. Meta-evaluation experiments revealed that scores output by VisCE$^2$ have excellent consistency with human judgments, especially in caption accuracy, which outperforms existing evaluation metrics. Future work includes providing VLM-based automatic evaluation along various perspectives, which would provide even more fine-grained information for humans.

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

## A  DATASET DETAILS

Here we provide the detail information of the dataset used in the experiments (Sec. 4).

**THumB 1.0** (Kasai et al., 2022). consists of 500 images sourced from MSCOCO, each assigned one human-written caption and four automatic-generated captions. The evaluation of candidate captions involves manual assessment based on three criteria: precision (how precisely the caption describes the image), recall (how well the caption covers the salient information in the image), and total (the overall quality of the caption, including fluency, inclusive language, and conciseness).

**Flickr8k-Expert** (Hodosh et al., 2013). contains 5,644 pairs of images collected from Flickr and automatically generated captions, each evaluated by three experts. A score of 1 indicates that the caption is unrelated to the image, and a score of 4 indicates that it accurately describes the image.

**Composite** (Aditya et al., 2015). includes 2,007 images sourced from MSCOCO (Chen et al., 2015), 997 images from Flickr8k (Hodosh et al., 2013), and 991 images from Flickr30K (Young et al., 2014). Each image is assigned two automatically generated captions and one human-written caption. Candidate captions are evaluated on a scale from 1 (irrelevant) to 5 (ideally related).

**Pascal-50S** (Vedantam et al., 2015). comprises 4,000 images from the UIUC Pascal sentence dataset (Rashtchian et al., 2010), each paired with candidate captions: one written by a human and the others automatically generated using five different methods. Annotators were asked to determine which caption was more similar to the reference.

## B  PROMPTS

We provide the prompts used in our experiments in Table 4.

## C  MODEL DETAILS

We present models used in the experiments. All models are publicly available in the huggingface, liuhaotian/llava-v1.5-7b, liuhaotian/llava-v1.5-13b, liuhaotian/llava-v1.6-vicuna-7b, liuhaotian/llava-v1.6-vicuna-13b, and MAGAer13/mplug-owl2-llama2-7b.

**LLaVA-v1.5** (Liu et al., 2023a). is an advanced large vision and language model designed to integrate visual and textual data through visual instruction tuning. Building on its predecessor, LLaVA, this model features a fully-connected vision-language connector using `CLIP-ViT-L-336px` as the vision encoder and an MLP projection layer, which significantly enhances data efficiency and performance. It outperforms other open models on visual reasoning and instruction-following capabilities.

**LLaVA-v1.6 (LLaVA-Next)** (Liu et al., 2024). Compared with LLaVA-1.5, LLaVA-NeXT has increased the input image resolution to 4x more pixels.

**mPLUG-Owl** (Ye et al., 2023). is a novel multi-modal language model designed to integrate visual and textual data through a modularized training paradigm. This model leverages a foundation LLM, a visual knowledge module, and a visual abstractor module to enable robust alignment between images and text. Utilizing a two-stage training approach, mPLUG-Owl aligns visual and textual information effectively.

## D  ADDITIONAL ANALYSIS

### D.1  PERFORMANCE OF GPT-4V

Before the meta-evaluation of GPT-4o, we experimentally investigated the evaluation performance using GPT-4V. GPT-4 family exhibits superior performance on many vision and language tasks (Yang et al., 2023), and it has also been quantitatively validated to outperform LLaVA-v1.5 in several evaluation tasks (Li et al., 2024; Cui et al., 2023). Due to budgetary constraints, we selected the relatively small yet finely-annotated dataset, THumB to compare LLaVA-v1.5-13B

Table 4: The prompts used in the experiment. {caption}, {references}, and {context} indicate the place to insert. The image is given at the beginning of each prompt.

| Method | Prompt |
|---|---|
| Vanilla | On a precise scale from 0 to 100, rate whether the candidate caption is appropriate for the given image.
Candidate caption: {caption}
Your rating must be a single digit between 0 and 100. |
| w/ Reference | On a precise scale from 0 to 100, rate whether the candidate caption is appropriate for the given image.
Candidate caption: {caption}
Use the image and the following reference to evaluate the candidate caption:
Reference: {references}
Your final rating must be a single digit between 0 and 100. |
| w/ Description
Step 1. Context Extraction | Generate a detailed description for the given image. |
| **VisCE$^2$**
Step 1. Context Extraction | Analyze the uploaded image and provide a structured output focusing on the objects, their features, and the relationships between them. Select up to five of the most important elements. The output should be organized as follows:
List of Important Objects (up to five):
- Object 1: [Brief description]
- Object 2: [Brief description]
- (Continue as necessary, up to five objects)
Features (Specific characteristics and attributes of each object, such as color, shape, size, and texture):
- Features of Object 1: [Detailed description of features]
- Features of Object 2: [Detailed description of features]
- (Continue as necessary for each selected object)
Relationships (The way objects interact or are positioned relative to each other, without using specific object names or symbols):
- Description of a relationship: [General description]
- Another relationship: [General description]
- (Continue as necessary for each relevant relationship)
Focus on providing unique and detailed insights into the features and relationships of the selected objects up to five objects. |
| w/ Description
Step 2. Evaluation

**VisCE$^2$**
Step 2. Evaluation | On a precise scale from 0 to 100, rate whether the candidate caption is appropriate for the given image.
Candidate caption: {caption}
Use the image and following visual context to evaluate the candidate caption:
Visual context: {context}
Your final rating must be a single digit between 0 and 100. |

with `gpt-4-vision-preview` via the Azure OpenAI API, using the different visual context resources (Vanilla and VisCE$^2$ setting.)

As shown in Table 5, GPT-4V outperformed LLaVA-v1.5 for precision and total score at both prompt settings. Similar to the LLaVA-v1.5 model, applying VisCE$^2$ to GPT-4 enhances the evaluation performance, indicating that the proposed method is effective even with more advanced VLMs. In addition, GPT-4V with VisCE$^2$ achieved state-of-the-art performance in terms of correlations with precision and total score. Furthermore, the evaluation performance of the LLaVA-v1.5 model with VisCE$^2$ applied was improved to the same extent as the performance of GPT-4V in the Vanilla setting. The effectiveness of our auto-evaluation method using this open-source

### D.2 ADDITIONAL QUALITATIVE EXAMPLES

We list several additional qualitative examples in Table 6 to further clarify the effectiveness of our VisCE$^2$. The examples are drawn from the THumB, Flickr8k-Expert, Composite, and THumB. Each

Table 5: Comparison VLM-based caption evaluation methods on correlation (Pearson's $\rho$) with human judgement on THumB 1.0.

| VLM | Method | THumB w/o Pearson's $\rho$ | | | THumB w/ Pearson's $\rho$ | | |
|---|---|---|---|---|---|---|---|
| | | P | R | Total | P | R | Total |
| LLaVA-v1.5 | Vanilla | .44 | **.08** | .38 | .41 | **.08** | .34 |
| LLaVA-v1.5 | **VisCE**$^2$ | .54 (+.10) | **.08** ($\pm$0) | .45 (+.07) | .49 (+.08) | .07 (-.01) | .39 (+.05) |
| GPT-4V | Vanilla | .53 | .03 | .41 | .50 | .05 | .38 |
| GPT-4V | **VisCE**$^2$ | **.58** (+.05) | .06 (+.03) | **.46** (+.05) | **.55** (+.05) | **.08** (+.03) | **.44** (+.06) |

Table 6: Additional qualitative examples of comparison between evaluation scores of VisCE$^2$, that of CLIP-S, and human ratings for candidate caption for images.

| Image | Candidate Caption | Human | **VisCE**$^2$ | CLIP-S |
|---|---|---|---|---|
|  | a street sign on a pole in front of a building . | 5.0 | 75 | 0.26 |
| | person is casting light in the scene. location is showing group in the scene. group is walking. The scene contains street and people walk and booths and pavement. | 1.0 | 20 | 0.27 |
|  | a man is rock climbing . | 5.0 | 80 | 0.31 |
| | person is climbing set in the up.The scene contains old architecture structure and unfinished structures and buildings and palatial building and exterior. | 1.0 | 10 | 0.31 |
|  | a dog with its mouth opened. | 5.0 | 80 | 0.31 |
| | a brown dog with a white collar is licking its nose . | 1.0 | 50 | 0.31 |
|  | A white toilet sitting on the side of a street. | 3.5 | 50 | 0.38 |
| | A white toilet sitting on the side of a building. | 4.0 | 75 | 0.36 |
| | A toilet sitting outside a building in an alley. | 4.5 | 80 | 0.34 |
|  | A woman riding a paddle board in the water. | 3.5 | 50 | 0.34 |
| | A young boy riding a paddle board in a river. | 4.0 | 60 | 0.34 |
| | A man paddling a kayak down a river. | 5.0 | 80 | 0.33 |

example includes an image, two candidate captions and scores by human annotation, CLIP-S, and VisCE$^2$. CLIP-S fails to fine-grainly compare the longer sentence with short and concise sentence, and it tends to overestimate captions including the salient objects in the image. In contrast, VisCE$^2$ can accurately evaluate the quality of the captions by considering the detailed visual context of the image. The third one is another typical example where CLIP-S overestimates the caption containing errors despite the correct color combination. In contrast, VisCE$^2$ provides a certain score for the presence of a brown dog in the image, which is still consistent with human relative preference.

