# OpenReview forum: "Vision Language Model Based Caption Evaluation Method Leveraging Visual Context Extraction"
_ICLR.cc/2025/Conference — Submitted to ICLR 2025_

### Official Review · Reviewer_GyEp · 2024-10-31

**Soundness:** 3
**Presentation:** 3
**Contribution:** 2
**Rating:** 3
**Confidence:** 4

**Summary:**

This paper proposes a VLM-based image caption evaluation method called VisCE2. The proposed method first obtains structured visual context by prompting the VLM, and then evaluates candidate captions based on the extracted visual context and input image. Extensive evaluation experiments show that VisCE2 outputs scores that have good agreement with human judgment and outperform existing evaluation metrics.

**Strengths:**

1.	The paper is well-organized and clearly written.
2.	The proposed VisCE2 is intuitive. And evaluation experiments on multiple datasets demonstrate that the method outperforms existing evaluation metrics and meets human judgments.

**Weaknesses:**

1. The paper is somewhat weak on innovation. The method is simply based on two rounds of prompts, which makes the VLM automatically evaluate image captions, and its core is based on the in-context learning ability of the VLM. Assuming that only one round of prompt is used and combined with the chain-of-thought (CoT) method to make the VLM automatically mine the visual context, while setting the last sentence generated by the VLM as the evaluation result, can this also lead to a good image caption evaluation performance?

2. Since two rounds of prompts are required for the VLM to evaluating the image caption, resulting in a high time complexity of this evaluation method, which is not conducive to real-time evaluation. Can the authors provide a comparison of runtime with existing evaluation methods?

3. Based on Table 3 of the ablation experiment, the enhancement brought by visual context does not seem to be particularly significant compared to the original prompt (Vanilla). Can the authors further analyze the reasons for this condition?

**Questions:**

1. When simply constructing the initial prompt (Vanilla) in a more refined way, e.g. by adding a chain-of-thought (CoT) prompt, would better assessment results also be achieved?

2. Can the authors provide a comparison of runtime with existing evaluation methods?

3. The visual context extracted in the first phase will contain some hallucinations, does this have an impact on the evaluation results?

---

### Official Review · Reviewer_qq5v · 2024-11-03

**Soundness:** 2
**Presentation:** 3
**Contribution:** 2
**Rating:** 5
**Confidence:** 2

**Summary:**

The paper proposes a method that uses the visual concepts extracted by MLLMs to help evaluate image captions, which makes the evaluation results more consistent with human ratings.

**Strengths:**

The paper proposed a new method to evaluate the generated captions considering the objects, attributes, and relations within the images. And the paper makes great efforts to demonstrate the reliability of the evaluation method by comparing with human judgement. The results indicate this method is better consistent with human rating compared to other metrics.

**Weaknesses:**

The evaluation process heavily relies on the use of MLLMs in the following ways: 1. It utilizes MLLMs to extract visual concepts from images; 2. It employs MLLMs to generate evaluation scores for these image captions. If the candidate captions are generated by a same MLLM, the evaluation method may fail to provide a fair evaluation.

It seems that the evaluation time is significantly longer than the time required by other metrics, due to the use of MLLMs in two stages. How long does it take to evaluate one caption based on an image? Please provide concrete timing comparisons between the proposed method and existing metrics. Additionally, why is the image necessary in the Vision-LM Caption Evaluation stage? If the visual concepts are sufficient to represent an image, the evaluation could potentially be conducted without using the image, which might speed up the evaluation process. The paper should include an ablation study comparing performance with and without the image in the Vision-LM Caption Evaluation stage.

Also, the paper should add ablation studies on the used prompts, particularly regarding the maximum number of objects. According to the prompts shown in Table 4, the maximum number of objects extracted by MLLM is set to 5.  How could this choice affect the reliability of the evaluation method?

**Questions:**

Please see the questions in Weaknesses.

---

### Official Review · Reviewer_W9vw · 2024-11-03

**Soundness:** 2
**Presentation:** 3
**Contribution:** 2
**Rating:** 3
**Confidence:** 4

**Summary:**

Given the accelerating progress of vision and language modeling, accurate evaluation of machine-generated image captions remains critical.
In order to evaluate captions more closely to human preferences, metrics need to discriminate between captions of varying quality and content.
However, conventional metrics fall short of comparing beyond superficial matches of words or embedding similarities; thus, they still need improvement.
This paper presents VisCE2, a vision language model-based caption evaluation method.
The authors’ method focuses on visual context, which refers to the detailed content of images, including objects, attributes, and relationships.
By extracting and organizing them into a structured format, the authors replace the human-written references with visual contexts and help VLMs better understand the image, enhancing evaluation performance.

**Strengths:**

- The proposed method is easy to understand.
- The proposed method shows favorable performance compared to existing evaluation methods.

**Weaknesses:**

- The novelty of the proposed method is weak. The only idea in this paper is to use a language model instead of CLIP to evaluate image captioning where the sentence generation performance of the VLMs is imperfect, unlike CLIP’s image-caption alignment performance. The authors suggest using an image captioning model to evaluate image captioning models. How can we evaluate the models that perform better than LLaVA? Using the proposed metric instead of CIDEr or CLIPS scores for future image captioning research is not convincing.

- The discussion on design choice is also weak. In Table 3, the only discussions are on what VLM to use and what kind of visual context to use. However, there are other design choices to be considered. For example, when using language models, a proper prompt is essential. However, the authors didn’t analyze the choice of prompts for the language model. Moreover, whether the visual context extractors (object, attribute, relation) have the best design choice isn't justified. Therefore, it is not clear whether the proposed metric is the best possible method.

- This paper lacks experimental analysis. When suggesting a new evaluation metric, it would be better to evaluate popular image captioning models, such as BLIP2, and analyze the tendency of the performances to understand the unique characteristics of the proposed metric. Also, it would be better to evaluate the proposed metric in different settings, such as FOIL hallucination detection, as CLIPS did.

**Questions:**

Please refer to the questions in the weakness.

---

### Official Review · Reviewer_eDjE · 2024-11-04

**Soundness:** 3
**Presentation:** 3
**Contribution:** 2
**Rating:** 3
**Confidence:** 4

**Summary:**

This paper introduces a reference-free image captioning evaluation metric, called VisCE$^2$. Specifically, VisCE$^2$ leverages pre-trained Vision-Language models (VLMs) to realize two-stage measurements for candidate captions. The first is Visual Context Extraction which uses VLM to obtain detailed descriptions including objects, object attributes and relationships. The second is Vision-LM Caption Evaluation which takes visual context, image and candidate captions as inputs to obtain an evaluation score. Experimental results demonstrate the superiority of this reference-image free method against other metrics.

**Strengths:**

1. A novel reference-free image caption evaluation method with VLMs.
2. This paper is well-written and easy to follow.
4.  This paper proposes a visual context extraction module to describe the image as sentences, which also can be seen as a pseudo reference with abundant details.
4. The authors conduct comprehensive experiments across multiple datasets.

**Weaknesses:**

1. Figure 1 is not comprehensive. For the left part, RefCLIP-S[1] and RefPAC-S[2] can also accomplish the same measurement. On the other hand, better evaluation performances of VisCE$^2$ than BLEU-4, ROUGE, SPICE and CIDEr are not enough. While for the right part, authors should compare with PAC-S[2] to illustrate the superiority of this work.

2. Line 49 - Line 51 describes the disadvantages about InfoMetIC, but evidence is lacked and can therefore be listed in Figure 1.

3. It is suggested to evaluate the VisCE$^2$ and other reference-free metrics within different *image captioning methods* such as InstructBLIP, LLaVA and even GPT-4, as mentioned in Line 42-Line 44. This is a key step to comprehensively measure the effectiveness of VisCE$^2$. The authors can refer to Table 7 in PAC-S paper[2].

4.  Although this paper focuses on reference-free evaluation, it is also recommended to report the results of VisCE$^2$ when the reference captions are provided.

5.  An example of visual context given the image should be added into appendix. For instance, authors can list all the objects, object attributes and relationships about the image in Figure 2.

6. In Table 2, it seems that authors only report the values of Kendall’s $\tau_b$ on Flickr8k-Expert and Composite datasets. Kendall’s  $\tau_c$ should also be included.

7. It is a little bit confusing to read Table 3 about ablation experiments. The first two settings are to prove the effectiveness of each component with the same backbone VLM (LLaVA-v1.5-13B). Then the current model (VisCE$^2$ ours) achieves the best scores across all datasets. But for the last two settings, authors aim to explore the influences of different backbone models or model sizes. From Table 3, GPT-4o can achieve **59.0** score on Composite dataset, higher than VisCE$^2$(**56.0**). THumB and Pascal-50S observe similar phenomenon.  Hence, it would be better to split Table 3 into two small tables.

[1] Hessel, Jack, et al. "Clipscore: A reference-free evaluation metric for image captioning." arXiv preprint arXiv:2104.08718 (2021).

[2] Sarto, Sara, et al. "Positive-augmented contrastive learning for image and video captioning evaluation." Proceedings of the IEEE/CVF conference on computer vision and pattern recognition. 2023.

**Questions:**

Please see weaknesses.

I will be happy to raise my score if authors address my concerns.

---

### Official Review · Reviewer_mpwh · 2024-11-08

**Soundness:** 2
**Presentation:** 3
**Contribution:** 2
**Rating:** 3
**Confidence:** 4

**Summary:**

The paper presents VisCE2, a vision-language model-based caption evaluation method designed to evaluate captions in a manner that aligns more closely with human preferences. VisCE2 focuses on visual context, which refers to the detailed content of images, including objects, attributes, and relationships. Experiments are conducted on several datasets.

**Strengths:**

- The paper highlights the urgent need for developing new metrics, considering the fact that model generations have become so detailed that they often exceed the capability of the automatic evaluation metrics.

- The paper is easy to follow.

**Weaknesses:**

- The literature review should be more accurate. For example, SPICE (Anderson et al., 2016) is mainly based on scene graphs rather than n-grams.

- The novelty of this paper is limited. The proposed evaluation method consists two stages: visual context extraction and VLM-based caption evaluation. The first stage analyzes images based on scene graphs, similar to SPICE (Anderson et al., 2016).  The second stage evaluates captions with VLMs, which is not new given existing works such as InfoMetIC (Hu et al., 2023) and CLIPScore (Hessel et al., 2021). While  the combination of these two stages may be new, it may not meet the innovation standards expected for ICLR submissions.

**Questions:**

- What are the main differences between the proposed method and SPICE/InfoMetIC? What unique innovations does this paper offer?
- Why choose THumB instead of MSCOCO for evaluation?
- The meanings of the numbers should be stated more clearly. For example, what do 5/5 and 80/100 mean？

---

### Meta-Review · Area_Chair_HMqG · 2024-12-16

**Metareview:**

The paper introduces a vision-language model-based caption evaluation method.
Reviewers have raised major concerns on inaccurate literature review, limited novelty, insufficient comparisons, ablations, and analysis. All five reviewers recommended rejection. No rebuttal was provided.

**Additional Comments On Reviewer Discussion:**

Reviewers have raised major concerns on inaccurate literature review, limited novelty, insufficient comparisons, ablations, and analysis. All five reviewers recommended rejection. No rebuttal was provided.

---

### Decision · Program_Chairs · 2025-01-22

Reject